# Genome sequencing of *Plasmodium malariae* identifies continental segregation and mutations associated with reduced pyrimethamine susceptibility

Amy Ibrahim [1], Franziska Mohring[1], Emilia Manko[1], Donelly A. van Schalkwyk [1], Jody E. Phelan [1], Debbie Nolder[1,2], Steffen Borrmann[3], Ayola A. Adegnika[3], Silvia Maria Di Santi [4], Mohammad Shafiul Alam [5], Dinesh Mondal[5], Francois Nosten [6], Colin J. Sutherland [1,2], Robert W. Moon [1] ✉, Taane G. Clark [1,7] ✉ & Susana Campino [1] ✉

*Plasmodium malariae* parasites are widely observed across the tropics and sub-tropics. This slow-growing species, known to maintain chronic asymptomatic infections, has been associated with reduced antimalarial susceptibility. We analyse 251 *P. malariae* genomes from 28 countries, and leveraging 131,601 high-quality SNPs, demonstrate segregation of African and Asian isolates. Signals of recent evolutionary selection were identified in genes encoding putative surface proteins (*pmmsp1*) and putative erythrocyte invasion proteins (*pmdpap3, pmrbp2, pmnif4*). Amino acid substitutions were identified in orthologs of genes associated with antimalarial susceptibility including 2 amino acid substitutions in *pmdhfr* aligning with pyrimethamine resistance mutations in *P. falciparum*. Additionally, we characterise *pmdhfr* mutation F57L and demonstrate its involvement in reduced susceptibility to pyrimethamine in an in vitro parasite assay. We validate CRISPR-Cas9 mediated ortholog replacement in *P. knowlesi* parasites to determine the function of *pmdhfr* mutations and demonstrate that circulating *pmdhfr* genotypes are less susceptible to pyrimethamine.

*Plasmodium malariae* is a neglected causative agent of human malaria, with unique disease biology and an ability to chronically persist asymptomatically in the blood[1]. Whilst commonly asymptomatic, infections can lead to significant illness including anaemia, nephrotic syndrome[2] and splenomegaly[1]. *P. malariae* infections are difficult to diagnose, often presenting with low or sub-microscopic parasitaemias and frequently occurring as mixed infections alongside other Plasmodium species[3]. Molecular detection methods have revealed that *P. malariae* infections are more common than previously estimated, including over 50% of school children in Nigeria asymptomatically carrying the parasites[4]. *P. malariae* is frequently found on the African continent, and is also prevalent in South America, Asia, and Oceania[5–7].

[1]Faculty of Infectious & Tropical Diseases, London School of Hygiene & Tropical Medicine, London, UK. [2]Public Health England Malaria Reference Laboratory, London School of Hygiene & Tropical Medicine, London, UK. [3]Institute for Tropical Medicine, Eberhard Karls University of Tübingen, Tübingen, Germany; Centre de Recherches Médicales de Lambaréné, Gabon; and German Center for Infection Research (DZIF), Tübingen, Germany. [4]School of Medicine, University of São Paulo, São Paulo, Brazil. [5]International Centre for Diarrhoeal Disease Research Bangladesh, Mohakhali, Bangladesh. [6]Shoklo Malaria Research Unit, Mahidol-Oxford Tropical Medicine Research Unit, Faculty of Tropical Medicine, Mahidol University, Mae Sot, Tak, Thailand. [7]Faculty of Epidemiology & Population Health, LSHTM, London, UK. ✉e-mail: rob.moon@lshtm.ac.uk; taane.clark@lshtm.ac.uk; susana.campino@lshtm.ac.uk

Worryingly, current malaria control measures are likely to be less effective in targeting *P. malariae* parasites due to difficulties in diagnosing infections, documented reduced susceptibility to antimalarials[8,9] and chronic asymptomatic infections[1], all of which contribute to a hidden reservoir of infected individuals. Additionally, *P. malariae* may affect the outcome of *P. falciparum* in mixed infections, including increased gametocytaemia[10], reduced likelihood of fevers[11] and a reduced peak parasite load[12]. An improved understanding of the biology of *P. malariae* is essential to ensure that better treatments and control measures are available to assist the elimination of all human-infective Plasmodium species.

Whole genome studies of *P. falciparum* and *P. vivax* parasites have enabled investigations into population genetics and identified genomic regions under selective pressure, including genes linked to drug resistance[13–15]. Global genome datasets are a vital resource for biological interrogation which are lacking for the neglected Plasmodium parasite species, in part due to difficulties in obtaining sufficient DNA from low parasitemia infections. In this work, we create and analyse the largest database of whole genome sequences for *P. malariae* with isolates from 4 continents and investigate patterns of population structure and positive selection. We identify mutations in orthologs of genes associated with antimalarial susceptibility, and using ortholog replacement in *P. knowlesi* parasites, we describe circulating *pmdhfr* genotypes that are associated with reduced susceptibility to pyrimethamine.

## Results

### A global genomics database for *P. malariae*

Here, we harness selective whole genome amplification (SWGA)[16] to amplify and sequence whole genomes of 228 clinical isolates of *P. malariae*, de novo. Combining these novel genomes with 23 published sequences[5,16], we generate a genomics database of 251 isolates (Supplementary Data 1, 2, Supplementary Fig. 1) and identify 1,288,675 unique, genome-wide SNPs in comparison to the reference strain (PmUG01), an isolate originating from Uganda[5]. Subsequent stringent filtering generated a high-quality genomics database of 157 isolates spanning four continents and 28 countries (Africa n = 128; Algeria, Angola, Cameroon, Congo, Democratic Republic of Congo, Equatorial Guinea, Gabon, Ghana, Guinea, Ivory Coast, Kenya, Liberia, Malawi, Mali, Mozambique, Nigeria, Senegal, Sierra Leone, Sudan, Tanzania, Uganda. Asia n = 22; Bangladesh, Indonesia, Malaysia, Thailand. Oceania n = 1; Papua New Guinea. South America n = 6; Brazil, French Guiana) (Supplementary data 1, 3). The filtered database yielded 131,601 robust SNPs within the core genome (excluding hypervariable and subtelomeric regions)[16], carried forward for detailed population genomic analysis. To identify and validate candidate gene mutations, a more extensive sample set was used, which included the entire sequenced dataset, refined to encompass only isolates with a Centrifuge[17] score surpassing 0.9, removing isolates suggestive of multiple-species infections (resulting in an isolate count of 194, with a total of 1,152,341 SNPs).

### African and Asian *P. malariae* isolates are genetically distinct

Using 131,601 single nucleotide polymorphisms (SNPs), principal component (PC) and maximum likelihood (ML) analyses reveal distinct clustering of Asian and African isolates (Fig. 1, Supplementary Fig. 2). Within Africa, population structure appears complex, with no clear geographic clustering across the continent, consistent with previous studies on smaller genomic datasets[16] and microsatellite analyses[18] (Fig. 1A, B). The ML tree shows a close genetic relationship between African and South American isolates, resembling patterns observed in *P. falciparum*[19] (Fig. 1A). However, the South American isolates diverge from the African ones along PC2, suggesting some level of differentiation (Fig. 1B). Region-specific PCAs reveal clear country-level segregation among Asian isolates, while African isolates show no distinct population structure (Supplementary Fig. 2). There is a slight indication of separation between East and Southern African isolates versus those from Western and Central Africa along PC2, which could hint at regional structuring. However, this separation is not absolute, with numerous outliers suggesting a lack of consistent segregation by country (Supplementary Fig. 2). Further sequencing and analysis may clarify these patterns.

Ancestral relationships, investigated using ADMIXTURE software, revealed a similar separation between African and Asian isolates, with 2 ancestral populations most likely (K = 2, lowest cross-validation, CV error score) (Supplementary Fig. 3). Asian isolates consistently exhibited a single ancestry across different levels of ancestral designations (K = 2, 3, and 4). In contrast, higher resolution was observed in Africa at K = 4, indicating a preliminary segregation of African isolates, though this patterning is not the most likely with the current sample set (Supplementary Fig. 3). Together, the ML tree, PCA plots, and ADMIXTURE analyses demonstrate a clear distinction between Asian and African isolates (Fig. 1), showing parasite segregation at the continental level. While the PCA and ADMIXTURE analyses suggest potential segregation within Africa, this pattern is neither clear nor statistically supported. When ancestral relationships within Africa were examined using four ancestral populations, regional differences emerged in the proportions of ancestries K1, K2, and K3. Broadly, West Africa displayed predominantly K1 and K2 ancestries, Central Africa with higher levels of K2 and K3, and East and Southern Africa with a majority of K2 (Fig. 1C). It is unclear with this sample set, whether these are true patterns of segregation, demonstrating the need for more sequence data for African *P. malariae* parasites. South American parasites demonstrate an African ancestry with K = 2 ancestral populations (Fig. 1C), with part shared ancestry with Asian isolates when visualised with K = 4 ancestral populations. As the most likely number of ancestral populations is 2, this ADMIXTURE analysis can confidently infer that African isolates are distinct from Asian isolates, with South American isolates demonstrating shared ancestry with isolates from Africa.

### Geographically variant residues within surface, invasion, and transmission associated proteins

A total of 167 completely differentiating SNPs (84 of which are missense mutations) driving population separation between African and Asian isolates were identified using the fixation index, $F_{ST}$ ($F_{ST} = 1$) (Supplementary data 4). Missense differentiating SNPs were identified in putative AP2 transcriptional factors (PmUG01_09017400 1084 P > 1084S and PmUG01_14056700 154H > 154Q), surface proteins including the GPI-anchored micronemal antigen (*pmgama* 254 L > 254 M), merozoite surface proteins (*pmmsp9* 29 T > 29 A, 425 V > 425D; PmUG01_04025700 235 N > 235 K), and the within the C-terminal region of the circumsporozoite surface protein (*pmcsp* 283 K > 283Q, 295E > 295 G), an ortholog of a *P. falciparum* parasite antigen, where the NANP repeat, and C-terminal region is used in the development of the only approved malaria vaccine, Mosquirix[20]. There were also differentiating SNPs in genes involved in erythrocyte invasion including the rhoptry neck protein *pmron2*[21] (1728 R > 1728 K) and *pmripr* (1018D > 1018 N, 1109 R > 1109 G). *Pfripr*, the *P. falciparum* ortholog forms part of a complex that binds the human erythrocyte receptor, basigin[22]. Additionally, differentiating SNPs were identified in genes associated with mosquito development including the cystine repeat modular protein (*pmcrmp* 1963S > 1963P), associated in *P. berghei* with host cell targeting within the mosquito salivary gland[23], and an essential transcription factor for gametocytogenesis, *pmap2-g* (1916S > 1916R)[24] (Supplementary data 4). Regional differences in genes associated with mosquito development may reflect distinct co-evolutionary selection pressures related to locally prevalent Anopheles species in different geographical locations, however functional roles are based on orthologues in *P. falciparum* and *P. berghei* and these genes may have distinct roles in *P. malariae*.

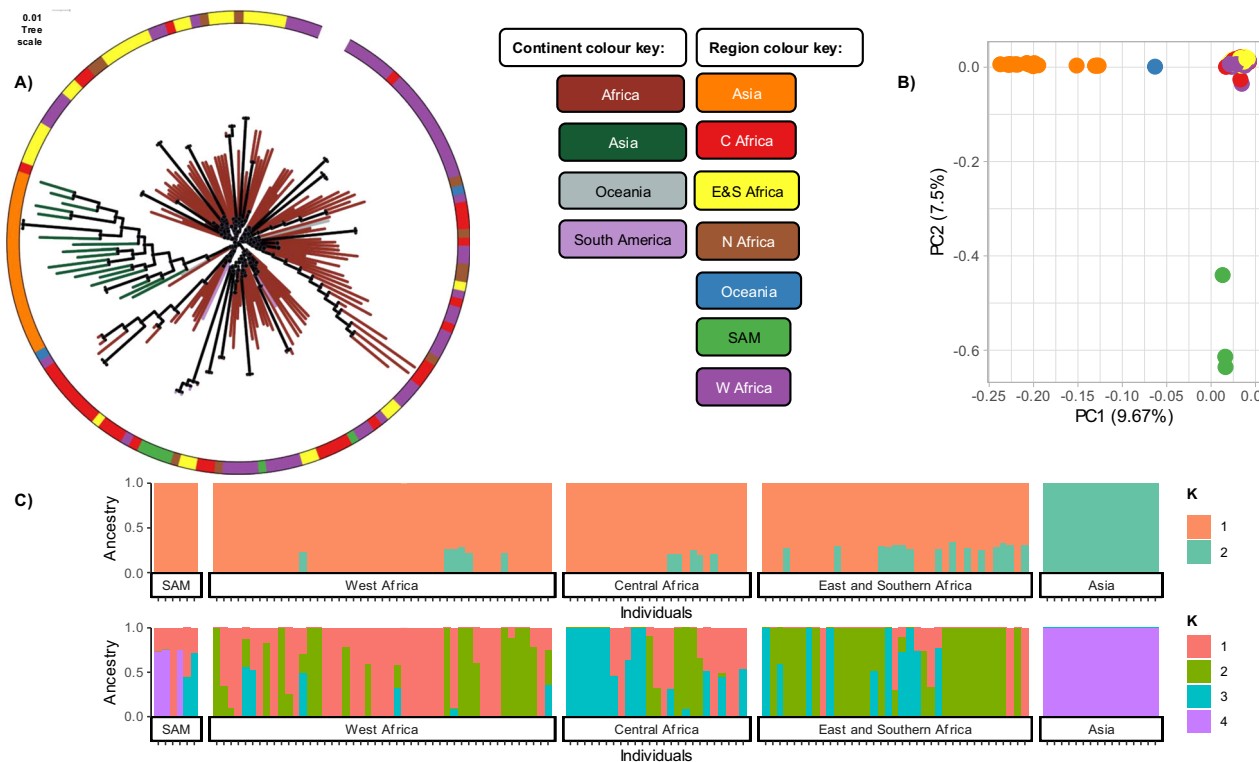

**Fig. 1 | Population structure of *P. malariae* isolates (n = 157) using 131,601 high-quality filtered SNPs. A** Maximum-likelihood tree calculated using IQTREE with ModelFinder (GTR + F + R4 model), ultrafast bootstrap and sh-aLRT test both to 1000 replicates. Bootstrap scores >50 annotated with a black circle, and branches coloured by isolate continent. The region (separating the African continent into subregions) is annotated using the outer circular track where each isolate has a corresponding bar on the outer circle coloured to the region of origin. C Africa = Central Africa, E&S Africa = Eastern and Southern Africa, N Africa = Northern Africa, SAM = South America, W Africa = Western Africa. **B** Principal component analysis (PCA) of 157 isolates based on a Manhattan distance matrix, isolates coloured by region. **C** Admixture prediction of ancestral populations (K) per sample, visualised using 2 ancestral populations, which had the lowest cross-validation (CV) error score, and using 4 ancestral populations. The CV error score for K = 4 was higher, however this is used only to visualise preliminary segregation of African isolates and is not the most likely ancestral pattern. Isolates ordered according to the country of origin and grouped into regional population groups. Admixture analysis used only with countries that had > 5 isolates, (141 total isolates). Each bar represents an isolate which is coloured to reflect the percentage accordance with each predicted ancestral population.

## Inbreeding rates across *P. malariae* isolates

Within-host parasite diversity was investigated using the inbreeding coefficient metric ($F_{WS}$)[25], and revealed lower levels of diversity (higher $F_{WS}$ value) in African and South American parasites (mean $F_{WS}$ = 0.956 and 0.959 respectively), with higher levels in Asia (mean $F_{WS}$ = 0.903). A Wilcoxon test revealed evidence of a difference in $F_{WS}$ between African and Asian isolates ($P = 0.0043$), but not between African and South American ($P = 0.33$). Overall, most filtered isolates (136/157; 86.6%) had a $F_{WS}$ score > 0.85 (Africa 115/128, Asia 16/22, South America 5/6) (Supplementary Fig. 4) indicating monoclonal infections.

## Global signals of homology within genes associated with parasite development, erythrocyte invasion and drug susceptibility

Monoclonal isolates ($n = 136$) were used in identity by descent (IBD) analyses of genetic relatedness to identify genomic regions with shared ancestry. South America ($n = 5$) demonstrated the most homogenous parasite population (mean IBD: 0.133, range: 0.014–0.470), consistent with most isolates being sourced from a small region on the southeast coast of Brazil. Asian isolates ($n = 16$, mean: 0.105, range: 0.006–0.948) had higher IBD than African isolates ($n = 115$, mean: 0.037, range: 0-1), which were more heterogenous (Supplementary Fig. 5, Supplementary data 5). The distribution of pairwise IBD scores (comparing isolates within each continent) highlighted significant differences between continents. African isolates exhibited a wide range of IBD scores, with the majority (87%) falling within the 0–0.05 range. In contrast, in Asia, 42% of pairwise comparisons had IBD scores between 0.05 and 0.1, while only 25% fell

within the 0–0.05 range. (Supplementary data 5). Scanning the genome in 10 kbp sliding windows identified regions of high IBD in Africa, Asia, and South America on chromosome 10, which included a bromodomain protein (*pmbdp1*) and an acetyl-coA transporter gene (*pmact*) (Supplementary Fig. 6, Supplementary data 6). *Pmbdp1* is an orthologue of a chromatin binding protein which silences expression of genes involved in erythrocyte invasion in *P. falciparum*[26], and *pmact* is a transporter whose ortholog has been implicated in *P. falciparum* multi-drug resistance[27]. South American signals of homology included two regions in chromosome 5 (IBD > 0.2) encompassing a gene associated with both asexual and sporozoite development (*pmlipb*)[28], a signal within chromosome 7 (IBD = 0.28) localising with an ATP-dependant protease subunit essential for parasite growth and development (*pmclpy* gene)[29], and two regions of homology within chromosome 12 (IBD = 0.3) encompassing a gene encoding an ABC transporter (*pmabcb6*), potentially involved in drug resistance[30] (Supplementary Fig. 6, Supplementary data 6).

## African signals of recent positive selection in proteins associated with erythrocyte binding and invasion

Signals of recent positive selection within populations were investigated using the integrated haplotype score (iHS) for each continent (Africa: 155 monoclonal isolates; Asia: 16 monoclonal isolates), applying the established REHH approach[31]. Additionally, differential selection signals between African and Asian isolates were assessed using Rsb values, also through the REHH approach. African signals of recent positive selection (iHS P-value < 10^{-4}) were observed in genes encoding

the merozoite surface protein (*pmmsp1*) in addition to signals associated with calcium signalling (parasite kinase *pmpip5k*[32]), processing of parasite invasion ligands (dipeptidyl aminoprotease, *pmdpap3*[33]) and the nuclear protein phosphatase (*pmnif4*) whose ortholog in *P. falciparum* has been associated with merozoite invasion and artemisinin susceptibility[34] (Supplementary data 7). Asian signals of selection were observed within Plasmodium genes of unknown function, a hypothetical merozoite protein (PmUG01_10046700), a ring finger protein (*pmrnf1*), and an ATP-dependant RNA helicase (*pmdbp6*) (Supplementary data 7). Differential signals of recent positive selection (Rsb P-value < 10$^{-5}$) between African and Asian populations were identified in genes encoding a hypothetical protein (PmUG01_10046500) and a conserved protein of unknown function (PmUG01_13011300). These signals also included *pmlisp2*, a protein involved in liver-stage parasite development[35,36], and a merozoite surface protein-like gene (PmUG01_12030100) (Supplementary data 8). Although orthologous genes in other Plasmodium species may suggest potential functions for *P. malariae* genes, their specific roles in *P. malariae* require functional validation.

### Circulating *P. malariae* parasites demonstrate variants in drug susceptibility-associated genes

There are no confirmed molecular markers for drug resistance in *P. malariae*, but treatment failures have been recorded[8,9], and in coinfections, parasites are frequently exposed to drugs used for treatment of the other species present. We screened for SNPs in candidate drug resistance genes (Supplementary data 9) and identified 291 missense variants across 18 loci (Supplementary data 10). High frequency SNPs (> 10% in single population or globally) include geographically fixed SNPs within the multi-drug resistance protein, *pmmdr2* (1028 V > 1028 A; fixed globally), the ubiquitin hydrolase, *pmubp1* (2307I > 2307S; fixed in Asia and Africa), and the gene encoding the dihydrofolate reductase enzyme, *pmdhfr-ts* (58 R > 58S; fixed in Asia and South America). *Pmmdr2* is a putative ABC transporter whose *P. falciparum* ortholog is associated with susceptibility across multiple antimalarial drugs, including artemisinin resistance due to amino acid substitution T484I[37] (Supplementary data 10). Amino acid substitutions in *P. falciparum* (V3275F[38]) and *P. chabaudi* (V739F, V770F[39]) orthologs of *pmubp1* are implicated in reduced susceptibility to artemisinin, with *P. falciparum* *pfubp1* variants associated with resistance and strong positive selection[40]. The *dhfr-ts* gene encodes an essential bifunctional enzyme in *P. falciparum* involved in folate biosynthesis that can be inhibited by pyrimethamine, with pyrimethamine resistance in *P. falciparum* due to amino acid substitutions encoded in *pfdhfr-ts*[41].

Other frequent missense SNPs were identified in genes associated with susceptibility to chloroquine (chloroquine resistance transporter *pmcrt*[42] and amino acid transporter *pmaat1*[43]), artemisinin (*pmubp1*[38,39], Mu subunit of clathrin-associated adaptor protein 2 *pmap2mu*[39]), lumefantrine and mefloquine (autophagy protein *pmatg11*[44]), pyrimethamine (*pmdhfr-ts*[45]), sulfadoxine (dihydropteroate synthase *pmdhps*[46]), genes encoding transporter proteins involved in multidrug responses (multi-resistance proteins *pmmrp1* and *pmmrp2*[47]), ABC transporters (*pmmdr1* and *pmmdr2*[48]) and the cation ATPase *pmatp4*, whose ortholog in *P. falciparum* is a target for several novel antimalarial compounds[49,50], with ortholog replacement in *P. knowlesi* demonstrating that this gene is associated with differing susceptibility between parasite species[51] (Supplementary data 10).

### In silico tertiary protein structures reveal *P. malariae* DHFR mutations align with validated resistance markers in *P. falciparum*

For established *P. falciparum* amino acid resistance markers, namely in AAT1[43], CRT[42], DHFR[41], DHPS[46] and K13[52] proteins, we sought to identify analogous sequence alterations in *P. malariae*. There were six low

frequency amino acid substitutions in PmK13 (Asia: M156V; Africa: Y372H, V404I, T505V, F507Y, V625I) (Supplementary data 10), whose ortholog in *P. falciparum* has been associated with artemisinin resistance[52]. No *pmk13* variants found in this study aligned with validated resistance markers in *pfk13* (Supplementary Fig. 7). Additionally, amino acid substitutions within PmCRT and PmAAT1 were not localised with validated resistance markers in the *P. falciparum* ortholog[42,43]. Five amino acid variants within PmDHFR (A15S, S49R, F57L, R58S and N114S) align closely with four mutations associated with drug susceptibility in PfDHFR (A16V, N51I, C59R and S108N[41]) (Supplementary Fig. 7), with four of these PmDHFR variants at high frequency in at least one population (A15S, F57L, R58S and N114S) (Supplementary data 10, 11, Supplementary Fig. 7). Two of the seven amino acids altered within PmDHPS align with sulfadoxine resistance mutations in the orthologous PfDHPS protein (Supplementary Fig. 7, Supplementary data 10, 11), highlighting both PmDHFR and PmDHPS as candidates for further functional investigation. However, due to the lack of a standardised in vitro sulfadoxine susceptibility assay[53], PmDHPS is not investigated further in this study, instead focussing on the functional validation of PmDHFR mutations and their effect on pyrimethamine susceptibility.

The *pmdhfr-ts* gene encodes three protein domains (DHFR, a linker domain and thymidylate synthase - TS), with only mutations leading to amino acid substitutions in the DHFR domain of the encoded PfDHFR protein validated as pyrimethamine resistance markers (Fig. 2). Two of the PmDHFR variant amino acids (58 and 114) overlap exactly at the secondary and tertiary structure level with *P. falciparum* positions (C59R and S108N) associated with pyrimethamine resistance[41] (Fig. 2). In *P. falciparum* (PfDHFR), mutations at positions 59 (arginine, 59R) and 108 (asparagine, 108N) are associated with pyrimethamine resistance. These correspond to identical amino acids at positions 58 (R58) and 114 (N114) in the *P. malariae* DHFR reference sequence (PmUG01_05034700), suggesting reduced pyrimethamine susceptibility in the *P. malariae* reference strain (PmUG01). This strain was derived from a clinical isolate of a traveller returning to Australia from Uganda[5] (Fig. 2). The amino acids variant from the reference genome (58S and 114S) are hypothesised in this study to be associated with a pyrimethamine-sensitive phenotype, (with the matching S108 in PfDHFR corresponding to pyrimethamine susceptibility). The F57L PmDHFR substitution, whilst not a validated pyrimethamine resistance marker within *P. falciparum*, has an orthologous 57L variant within *P. vivax* (PvDHFR). The PvDHFR F57L mutation is found in high frequency in Indonesia and Papua New Guinea, with reduced pyrimethamine susceptibility demonstrated in yeast[54]. We identified the *P. malariae* 57L variant in Africa, where it was most prevalent in Central Africa (49%) and East and Southern Africa (47%), and less frequent in West Africa (10%) (Supplementary data 11). The 58S and 114S variants were present in high frequency in all populations (Supplementary data 11). In Africa, the putative *P. malariae* resistant alleles (R58 and N114) were present in West Africa (both 20%), East and Southern Africa (25% and 20%, respectively) and in Central Africa (51% and 36%, respectively) (Supplementary data 11). The N114 variant was also observed in Asia (16%), where both F57 and 58S were fixed. The A15S substitution in PmDHFR aligns with PfDHFR position 16, associated with cycloguanil resistance[55] and thus, was not further investigated within this study (Supplementary data 11).

### Ortholog replacement in *P. knowlesi* validates that circulating PmDHFR variants are less susceptible to pyrimethamine

To assess the phenotype of amino acid alterations identified within PmDHFR, with the continued lack of an in vitro culture protocol, we utilised ortholog replacement in the culture adapted *P. knowlesi* line[56,57]. Six transgenic lines of *P. knowlesi* were created, where the DHFR domain of PkDHFR was replaced with: A recodonised sequence of PkDHFR as a transfection control (Pkdhfr$^{OR}$), the sensitive PfDHFR

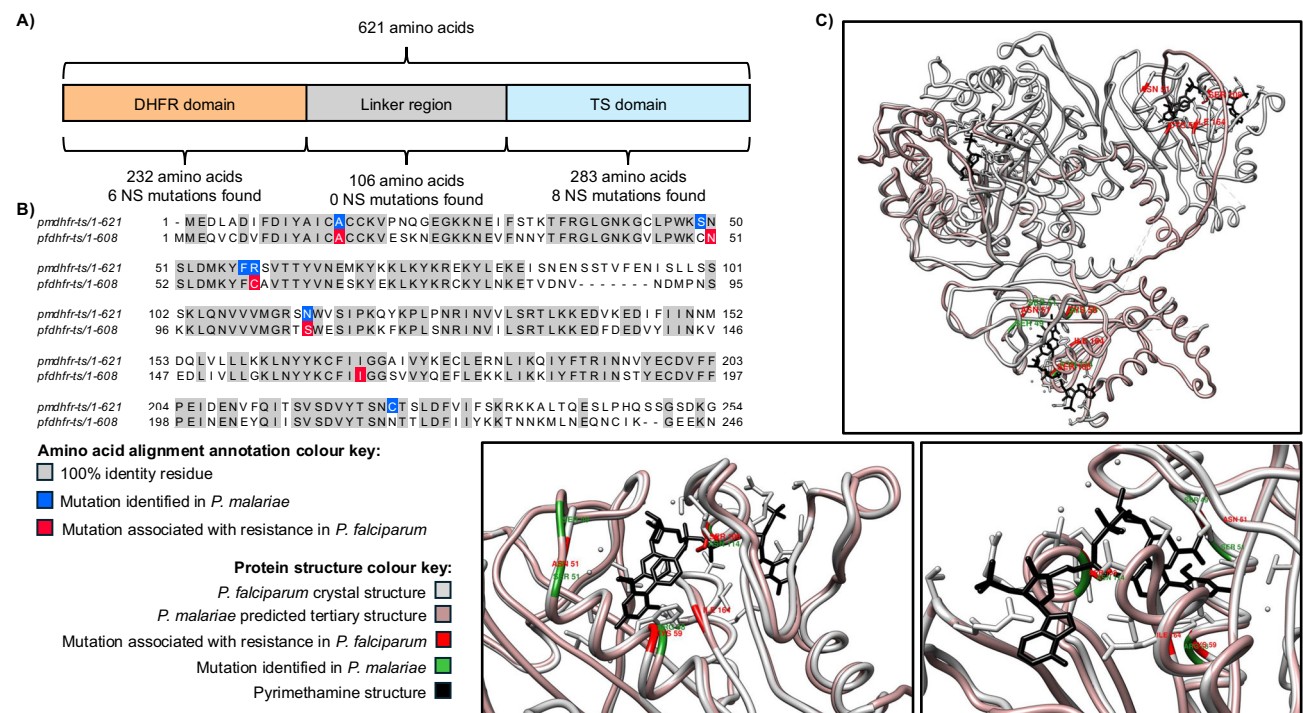

**Fig. 2 | Structure of DHFR and *pmdhfr-ts* variants. A** The amino acid sequence of *pmdhfr-ts* (PmUG01_05034700) was obtained from PlasmoDB[73] with domains annotated using Interpro domains (dihydrofolate reductase enzyme, DHFR: IPR001796; thymidylate synthase, TS: IPR000398). The number of nonsynonymous mutations (NS) found in each domain in the *P. malariae* database is annotated. **B** Amino acid sequences were aligned using Clustalo[74] and visualised in Jalview[75]. Conserved amino acids between *P. malariae* and *P. falciparum* highlighted in grey, with mutations identified in *P. malariae* highlighted in blue and amino acid positions associated with pyrimethamine resistance in *P. falciparum* highlighted in red. **C** The crystal structure of PfDHFR-TS (with pyrimethamine bound, 3QGT) was downloaded from RCSB Protein Data Bank (grey) and the structure of PmDHFR-TS was predicted using I-TASSER[76] (pink). Protein structures were aligned and visualised in UCSF Chimera[77], highlighting the overlapping mutations around the pyrimethamine binding site, with the structure for pyrimethamine in black. *P. malariae* variant positions are annotated in green (49, 57, 58, 114) and *P. falciparum* variants associated with pyrimethamine resistance are annotated in red (51, 59, 108, 164).

genotype with N51, C59 and S108 (Pfdhfr^OR_PfS), the resistant PfDHFR genotype with 51I, 59 R and 108 N (Pfdhfr^OR_PfR), the PmDHFR reference sequence including R58 and N114 (Pmdhfr^OR_PmUG01), a hypothetical sensitive PmDHFR genotype containing 58S and 114S substitutions (Pmdhfr^OR_PmS) and the reference PmDHFR genotype with the F57L mutation containing 57L, R58 and N114 (Pmdhfr^OR_PmM5).

Successfully integrated parasite lines were generated by co-transfection of two plasmids, with integration and generation of clonal lines validated by PCR (Supplementary Fig. 8). Previous Cas9-mediated genome editing in *P. knowlesi* required pyrimethamine as a positive selection drug, with human DHFR (hDHFR) acting as a resistance cassette in the Cas9 encoding plasmid[57]. In this study, we validate blasticidin as a positive selection drug for CRISPR-Cas9–mediated genome editing, allowing us to avoid pre-exposure of parasite lines to antifolates. Previous research successfully validated blasticidin as a selection marker using an episomally expressed resistance cassette, showing an IC50 of 20 mM for unmodified *P. knowlesi* parasites compared to 30 mM in modified lines carrying the episomal resistance cassette[58]. Growth assays confirmed that all six clonal transgenic parasite lines demonstrated no significant difference (Wilcoxon P > 0.05) in growth compared to the parental *P. knowlesi* A1-H.1 line (Pkdhfr^WT) (Supplementary Fig. 8).

Parasites were independently subject to serial dilutions of pyrimethamine and dihydroartemisinin (DHA) for one complete life cycle (27 h), with DHA as a control drug with a differing mechanism of action. These in vitro growth inhibition assays demonstrated contrasting susceptibilities to pyrimethamine and minimal difference in DHA susceptibility, with all lines demonstrating an $EC_{50}$ value below 7 nM for DHA (Fig. 3).

Pkdhfr^OR demonstrated no statistically significant variation to the parental PkA1-H.1 line (average $EC_{50}$ = 5 nM and 4 nM respectively; $P > 0.1$), confirming that transfection had minimal effect on the pyrimethamine phenotype (Fig. 3). All other parasite lines demonstrated significant differences in their pyrimethamine $EC_{50}$ values when compared to Pkdhfr^OR. Pfdhfr^OR_PfS was 5-fold less susceptible to pyrimethamine ($P < 0.001$, average $EC_{50}$ = 25 nM), consistent with previous observations[59]. Pmdhfr^OR_PmS, containing R58S and N114S variants, was 2.5-fold less susceptible to pyrimethamine than Pkdhfr^OR (average $EC_{50}$ = 13 nM), yet more susceptible than Pfdhfr^OR_PfS confirming pyrimethamine susceptibility. Differences in susceptibility between Pkdhfr^OR, Pfdhfr^OR_PfS and Pmdhfr^OR_PmS may reflect true species differences in drug susceptibility (Fig. 3). Pfdhfr^OR_PfR, containing 51I 59 R and 108 N, was 12099-fold less susceptible to pyrimethamine (average $EC_{50}$ = 64,148 nM ± 18,211; $P < 2 \times 10^{-15}$), as expected due to the high clinical resistance associated with this genotype. Pmdhfr^OR_PmUG01, containing R58 and N114, displayed a 940-fold reduced susceptibility to pyrimethamine (mean $EC_{50}$ = 4, 988 nM ± 2,430; $P < 2 \times 10^{-15}$), validating the involvement of PmDHFR amino acids R58 and/or N114 in reduced susceptibility to pyrimethamine (Figs. 2, 3). Pmdhfr^OR_PmM5, containing F57L on the background of a PmDHFR amino acid sequence, demonstrated a further reduction in pyrimethamine susceptibility (mean $EC_{50}$ = 28,264 nM ± 6644.7), 5331-fold less susceptible than Pkdhfr^OR ($P < 2 \times 10^{-15}$), which is the least susceptible *pmdhfr* genotype described (Fig. 3). Overall, our findings present compelling evidence of widely circulating *pmdhfr* genotypes associated with markedly reduced susceptibility to pyrimethamine (940- and 5331-fold reduced susceptibility), likely with clinical implications.

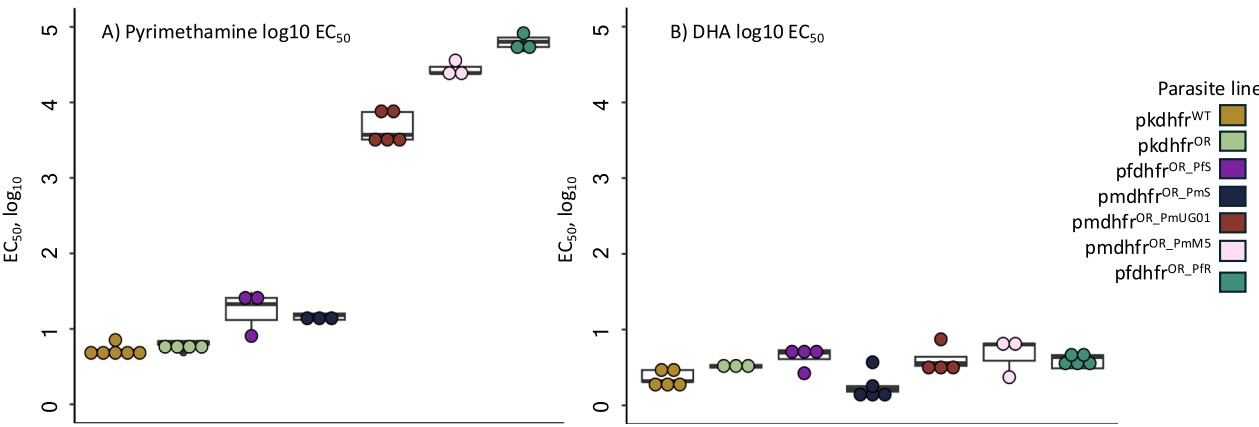

| Parasite line | A) Pyrimethamine EC$_{50}$ values | | | | B) DHA EC$_{50}$ values | |
|---|---|---|---|---|---|---|
| | Mean (nM) | SD | Fold change cf. Pkdhfr$^{OR}$ | Regression p-value cf. Pkdhfr$^{OR}$ | Mean | SD |
| Pkdhfr$^{WT}$ | 4.12 | 1.06 | NA | 0.197 | 1.34 | 0.56 |
| Pkdhfr$^{OR}$ | 5.30 | 0.93 | NA | NA | 2.30 | 0.27 |
| Pfdhfr$^{OR\_PfS}$ | 25.15 | 6.87 | 4.74 | <0.0001 | 3.54 | 1.30 |
| Pmdhfr$^{OR\_PmS}$ | 13.46 | 2.74 | 2.54 | 0.0008 | 0.97 | 1.00 |
| Pmdhfr$^{OR\_PmUG01}$ | 4988 | 2430 | 940 | <0.0001 | 3.35 | 2.10 |
| Pmdhfr$^{OR\_PmM5}$ | 28264 | 6644 | 5331 | <0.0001 | 4.11 | 2.39 |
| Pfdhfr$^{OR\_PfR}$ | 64148 | 18211 | 12099 | <0.0001 | 2.93 | 0.83 |

**Fig. 3 | Pyrimethamine susceptibility measured via growth inhibition assays of parasite lines.** Pkdhfr$^{WT}$ (Parental parasite line, yellow dot), Pkdhfr$^{OR}$ (recodonised *P. knowlesi* DHFR, transfection control line, green dot), Pfdhfr$^{OR\_PfS}$ (*P. falciparum* sensitive DHFR genotype: N51, C59, S108, purple dot), Pmdhfr$^{OR\_PmS}$ (*P. malariae* hypothetical sensitive sequence with two mutations: 58S and 114S, navy dot), Pmdhfr$^{OR\_PmUG01}$ (*P. malariae* reference sequence, with suspected pyrimethamine resistance: R58 and N114, red dot), Pmdhfr$^{OR\_PmM5}$ (*P. malariae* reference sequence: R58 and N114, with additional mutation F57L, pink dot), Pfdhfr$^{OR\_PfR}$ (*P. falciparum* resistant IRN sequence: 51I, 59 R and 108 N, aqua dot). All data points in both plots represent an independent biological replicate, with the overall mean EC50 in nM given in the summary table. For all lines, a minimum of 3 independent biological replicates were performed. For each independent biological replicate, 3 technical replicates were performed, and the mean EC$_{50}$ value across the technical replicates is visualised as an individual data point in each graph. In the table, the mean EC50 value across all the biological replicates is shown (Mean (nM)), with the standard deviation (SD) given. **A** Susceptibility to pyrimethamine using serial dilutions. A log$_{10}$ scale is used on the plot, with mean EC$_{50}$ values in nM in the table (standard deviations, SD). Regression analysis was used to investigate the differences in pyrimethamine susceptibility and asterisks used to demonstrate levels of significance ($p = 0 – 0.001$, ***; $p = 0.001 – 0.01$, **; $p = 0.01 – 0.1$, *, $p$ values). Regression analysis is a two-sided statistical test, and there is no adjustment for multiple comparisons as the testing is nested within the same model. Pkdhfr$^{WT}$ ($n = 6$

biological replicates, minima 0.629, maxima 0.851, median 0.673, Q1 0.653, Q3 0.734), Pkdhfr$^{OR}$ ($n = 4$, 0.691, maxima 0.836, median 0.828, Q1 0.792, Q3 0.831), Pfdhfr$^{OR\_PfS}$ ($n = 3$, minima 0.905, maxima 1.49, median 1.33, Q1 1.12, Q3 1.41), Pmdhfr$^{OR\_PmS}$ ($n = 3$, minima 1.06, maxima 1.23, median 1.18, Q1 1.12, Q3 1.20), Pmdhfr$^{OR\_PmUG01}$ ($n = 5$, minima 3.44, maxima 3.89, median 3.57, Q1 3.51, Q3 3.87), Pmdhfr$^{OR\_PmM5}$ ($n = 3$, minima 4.39, maxima 4.56, median 4.39, Q1 4.39, Q3 4.47), Pfdhfr$^{OR\_PfR}$ ($n = 3$, minima 4.66, maxima 4.92, median 4.80, Q1 4.73, Q3 4.86). *P*-values not included in the table include pfdhfr$^{OR\_PfS}$ 0.000768, pmdhfr$^{OR\_PmUG01}$ 2e-16, pmdhfr$^{OR\_PmM5}$ 2e-16, pfdhfr$^{OR\_PfR}$ 2e-16. **B** Dihydroartemisinin (DHA) was used as a control drug, with EC$_{50}$ values below 7 nM for all parasite lines. Pkdhfr$^{WT}$ ($n = 4$ biological replicates, minima 0.231, maxima 0.466, median 0.319, Q1 0.313, Q3 0.466), Pkdhfr$^{OR}$ ($n = 3$, minima 0.485, maxima 0.555, median 0.512, Q1 0.499, Q3 0. 533), Pfdhfr$^{OR\_PfS}$ ($n = 4$, minima 0.424, maxima 0.742, median 0.697, Q1 610, Q3 727), Pmdhfr$^{OR\_PmS}$ ($n = 5$, minima 0.071, maxima 0.569, median 0.214, Q1 0.180, Q3 0.254), Pmdhfr$^{OR\_PmUG01}$ ($n = 4$, minima 0.437, maxima 0.872, median 0.557, Q1 0.522, Q3 0.641), Pmdhfr$^{OR\_PmM5}$ ($n = 3$, minima 0.372, maxima 0.820, median 0.805, Q1 0.589, Q3 0.812), Pfdhfr$^{OR\_PfR}$ ($n = 5$, minima 0.474, maxima 0.668, median 0.638, Q1 0.489, Q3 0.638). In both assays (for pyrimethamine and DHA), pkdhfr$^{OR}$ is first compared to the parental parasite line (pkdhfr$^{WT}$) to confirm that the transfection process itself has no impact on drug susceptibility. Following this, all other parasite lines are compared to pkdhfr$^{OR}$ to determine the change in drug susceptibility.

## Discussion

The distinct biological features of *P. malariae*, such as its longer erythrocytic cycle and chronic asymptomatic blood-stage infections, pose significant challenges to disease elimination. Low-parasitaemia asymptomatic infections, which often go undetected and untreated, have been shown in *P. falciparum* to contribute substantially to malaria transmission[60]. One challenge for the development of infection control tools is poor understanding of the genomic diversity of *P. malariae* parasites. Genomics and functional studies were previously constrained by difficulties in whole genome sequencing (WGS) of low parasitaemia clinical isolates and the lack of an in vitro culture system respectively. Our study uses the previously validated SWGA methodology[16] to generate a large genomic dataset and Cas9-mediated orthologue replacement in *P. knowlesi* to validate the phenotypes of variants found within PmDHFR, identifying circulating genotypes with significantly reduced susceptibility to pyrimethamine.

Using core genome SNPs identified within 157 high quality genome sequences, we demonstrate genetic segregation between African and Asian isolates, with South American isolates aligning with those from Africa, supportive of an introduction of *P. malariae* into South America from an African origin as previously suggested[61]. SNPs driving differences between populations were present in genes encoding parasite surface proteins and mosquito stages, consistent with other studies[13]. ADMIXTURE analysis revealed a distinct Asian ancestral population, with a more complex population structure in Africa, with multiple ancestral populations and no clear regional clustering. Early population genetics studies of *P. falciparum* lacked the geographical resolution we can now achieve with over 7000 isolates[14]. For instance, a study with 227 isolates struggled to differentiate P. falciparum populations between Mali and Burkina Faso[25]. This suggests that increasing the number of sequenced *P. malariae* isolates in Africa could enhance geographic resolution for parasite clustering. However, the lack of clear segregation within Africa, even with over

100 sequenced isolates, may reflect a true unique characteristic of this species. Most isolates (87%) contained a major single genotype (mean $F_{WS}$ > 0.85). Multiplicity of infection (MOI) results may be skewed due to most isolates being obtained from returning travellers rather than endemic infections. However, a previous study demonstrated that MOI was comparable between isolates sourced from returning travellers and endemic infections, supporting the use of the former in genomic diversity studies[13].

Our understanding of drug resistance in *P. malariae* remains limited, with possible treatment failures recorded after artemisinin (combination)[8] and chloroquine treatments[9]. Global signals of genetic homology, suggestive of recent positive selection, encompassed several noteworthy genes associated with drug susceptibility, such as *pmact*, whose ortholog in *P. falciparum* has been associated with broad mechanisms of antimalarial resistance[27]. A detailed investigation of susceptibility-associated genes revealed missense SNPs within genes associated with susceptibility to chloroquine (*pmcrt* and *pmaat1*[42,43]), artemisinin (*pmubp1, pmap2mu*[38,39]), lumefantrine and mefloquine (*pmatg11*[44]) pyrimethamine (*pmdhfr*[41]), sulfadoxine (*pmdhps*[62]), novel antimalarial compounds (*pmatp4*[51]) and genes associated with susceptibility to multiple drug transport routes (*pmmrp1, pmmdr1, pmmdr2, pmmrp2*[30,47,48]), with high frequency SNPs found in *pmmdr2, pmubp1* and *pmdhfr-ts*.

We found that selected *P. malariae* mutations corresponded exactly to validated resistance markers in the *P. falciparum* and *P. vivax* orthologs, including mutations within PmDHFR, known to affect pyrimethamine susceptibility in *P. falciparum* and *P. vivax*. Using ortholog replacement within *P. knowlesi*, we validated the functional effects of PmDHFR genotypes found through WGS. The experimental approach using *P. knowlesi* ortholog replacement serves as a proof-of-principle for future investigations into genotype-phenotype associations across various drugs, facilitating functional studies in non-culturable Plasmodium parasites. Using this approach reveals that the reference genotype of PmDHFR (PmUG01_05034700) was markedly less susceptible to pyrimethamine, due to the arginine at position 58 and/or the asparagine at position 114. The reference genome for *P. malariae* (PmUG01) was sequenced from a clinical isolate obtained in Uganda, indicating the circulation of *P. malariae* parasites with reduced susceptibility to pyrimethamine in the region. The Pmdhfr[OR_M5] parasite line, containing the F57L mutation presented the greatest reduction in pyrimethamine susceptibility. The PmDHFR F57L mutation corresponds to position 57 in PvDHFR, associated with pyrimethamine resistance in yeast expression studies[54]. In this study, we validate using a parasite assay, the critical role of PmDHFR 57 L in in vitro pyrimethamine susceptibility, demonstrating a 5,331-fold reduction when this variant is combined with R58 and N114. Notably, PmDHFR 57 L is found exclusively in African *P. malariae* isolates. The 57 L amino acid, corresponding to position 58 in PfDHFR has not been reported as a resistance marker in this species. Further studies introducing this mutation into *P. falciparum* could determine its phenotype, potentially enabling the screening of highly resistant genotypes before they pose a clinical threat. Identifying novel PfDHFR resistance markers is particularly relevant in regions where artesunate-sulfadoxine-pyrimethamine (AS + SP) is used as an antimalarial treatment. Further investigation into the phenotypes of mutants orthologous to PmDHFR F57L in *P. falciparum* is essential. Through our global genomics analysis of *P. malariae*, we have identified PmDHFR variants that may have clinical significance due to their substantial reduction in pyrimethamine susceptibility observed in our ortholog replacement studies. Additionally, we propose potential novel drug susceptibility markers in *P. falciparum* for further investigation in areas experiencing delayed parasite clearance with AS + SP treatment.

All PmDHFR amino acid substitutions shown here to be associated with pyrimethamine susceptibility are common in African isolates (57L: 21%, R58: 22%, N114: 33%,) where *P. falciparum* pyrimethamine resistance is widespread. Beginning in the 1960s, pyrimethamine was globally adopted as a first-line antimalarial treatment in combination with sulfadoxine, collectively known as sulfadoxine-pyrimethamine (SP). Resistance to SP emerged rapidly in Southeast Asia by the 1970s and in East Africa by the late 1980s, resulting in the withdrawal of SP as a first-line therapy for malaria between 2003 and 2008. Despite this, SP remains in use for preventive treatment in many African countries, leading to ongoing exposure to pyrimethamine among African Plasmodium isolates compared to other regions[63]. Specifically, Central African isolates demonstrate the highest prevalence of all amino acids associated with reduced pyrimethamine susceptibility (57L; 49%, N114; 36%, R58; 51%) in comparison to isolates from West Africa and East and Southern Africa. Resistance associated variants 57L and R58 are not found outside of Africa, and N114 is only seen in Africa and Asia (16%; Asia). Asian and South American isolates exhibit fewer amino acid variants associated with reduced pyrimethamine susceptibility in PmDHFR, which may be related to the shorter exposure of Plasmodium parasites to pyrimethamine, as this drug is not used for chemoprevention in these regions. Cumulatively, this evidence strongly suggests that circulating *pmdhfr* mutations are linked to the selection of pyrimethamine resistance. In this study, *P. malariae* parasites from regions where pyrimethamine is still used show a higher prevalence of PmDHFR amino acid variants associated with reduced susceptibility. It is important to note that *P. malariae* infections are often asymptomatic and are rarely treated directly with antimalarials. Consequently, drug selection in *P. malariae* isolates is likely influenced by the use of pyrimethamine for treating other Plasmodium species, such as *P. falciparum*. This consideration is crucial for malaria control and underscores the potential impact of drug selection on other circulating species within populations.

Our study, the largest global WGS-based investigation of *P. malariae* to date, generated a comprehensive dataset crucial for guiding a targeted approach to investigate distinct genotypes, accurately representing the genetic diversity present in the natural population of *P. malariae*. Further large-scale investigations will provide greater resolution across malaria endemic populations, as well as temporal trends. We have provided an invaluable resource that can be exploited to understand the unique biology traits of the *P. malariae* parasite, as well as successfully implemented an orthologous gene replacement protocol to support functional studies, both ultimately informing tools to assist malaria control and elimination.

## Methods

### Ethics statement

Isolates from Thailand were obtained with ethical approval from the Mahidol Faculty of Tropical Medicine Ethics Committee (Ref: 2015-001.01). UKHSA Malaria Reference Laboratory samples were sourced with NHS UK Ethics approval (#18/LO/0738). All samples were collected according to in-country guidelines, and informed consent was obtained by all individuals and in the case of children, by their parents or legal representatives. All human blood used in this study for parasite culture is obtained though the UK National Blood Transfusion Service, where individuals have given informed consent for their blood to be used for medical research.

### Sample collection and processing

For this study, we used a total of 251 clinical isolates from *P. malariae* single-species infections including 228 isolates where we generated novel sequence data, and 23 previously published isolates. Of the previously published isolates, 4 were obtained in Malaysia ($n = 1$), Papua Indonesia (1), Guinea (1) and Mali (1)[5], and 19 were from Thailand ($n = 9$), Kenya ($n = 2$), Liberia ($n = 2$), Sudan ($n = 1$), Sierra Leone ($n = 2$) and Uganda ($n = 3$)[16].

Total samples were obtained from 31 countries covering 6 regions (South America (4 countries, $n = 16$), West Africa (10 countries, $n = 69$),

Central Africa (6 countries, $n = 55$), East and Southern Africa (6 countries, $n = 81$), Asia (4 countries, $n = 28$), and Oceania (1 country, $n = 1$). Additionally, one isolate was obtained with an unknown travel history, however, this isolate was filtered out of all analysis due to poor sequence quality (Supplementary data 1, 2). All isolates were collected between years 2001 to 2020. Blood samples obtained from the UKHSA Malaria Reference Laboratory were sourced from individuals who had previously returned to the UK with clinical symptoms of malaria after travelling to one known country with malaria transmission and were previously screened using both a nested PCR[64] and qPCR[65] for species identification. DNA for all samples was extracted from venous blood samples using the QIAamp DNA Blood Mini Kit (Qiagen) according to manufacturer's instructions. All DNA samples were screened to confirm single-species *P. malariae* infections using a qPCR assay, and the concentration quantified using a Qubit 3.0 fluorometer.

### Selective whole genome amplification and sequencing
All DNA samples underwent selective whole genome amplification (SWGA) to increase the relative levels of *P. malariae* DNA within the sample, using the primers: TATGTATA*T*T, TTATTC*G*T, TTCGTT*A*T, TTTTTA*C*G, TATTTC*G*T (asterisk denoting a phosphorothioate bond) as previously described[16]. Up to 60 ng of DNA was amplified in a 50 µl total reaction alongside 1X Phi29 DNA Polymerase Reaction Buffer (New England Biolabs), 1X BSA (New England Biolabs), 2.5 µM primer mix (Sigma), 1 mM dNTP (Roche), Nuclease-Free Water (Ambion, The RNA Company) and 30 units Phi29 DNA Polymerase (New England BioLabs). Reactions were carried out in a UV Cabinet for PCR Operations (UV-B-AR, Grant Bio) to minimise the risk of contamination. Amplified samples were purified using a 1:1 ratio of AMPure XP beads (Beckman-Coulter) following manufacturer's instructions.

For Illumina-based sequencing, libraries were prepared from clean DNA samples using either the QIAseq FX DNA Library Kit (QIAGEN) (with a 20-minute fragmentation step), or the NEB Next Ultra DNA Library Prep Kit (New England Biolabs, E7370). Sequencing on the Illumina platform (150 bp paired reads) was facilitated by The Applied Genomics Centre, LSHTM. DNA from four isolates sourced from Bangladesh (>200 ng) were sequenced using the Oxford Nanopore Technology (ONT) MinION platform (R9.4.1 flow cell; Ligation Sequencing Kit (SQK-LSK110), according to manufacturer's specifications at the LSHTM.

### Bioinformatic analysis
Illumina short read sequencing data (FASTQ format) was processed using TRIMMOMATIC software (v0.39) to remove poor quality ends of reads using the following parameters: LEADING:3 TRAILING:3 SLIDINGWINDOW:4:20 MINLEN:36. Trimmed reads were then mapped to the PmUG01 reference genome[5] using BWA-MEM software (v0.7.12). The resulting alignment (BAM) files were improved using the SAMTOOLS[66] (v1.9) functions fixmate and markdup before applying GATK's BaseRecalibrator and applyBQSR functions with a training set of high-quality SNP positions from previous work[16].

SNPs and indels were identified using GATK's HaplotypeCaller (v 4.1.4.1) following recommended best practice[67] with default parameters using the -ERC GVCF option, to produce two combined VCF files (one for SNPs, one for indels) of variants for all isolates. The set of variants was reduced to those only within the core genome (as previously described[16]). The combined SNP VCF file was further filtered to remove those with a Variant Quality Score Log-Odds (VQSLOD) less than zero, in addition to individual SNP positions with missing calls in >10% isolates, and individual samples with missing data at >30% of the SNP positions. Isolates were also filtered using Centrifuge software to check for contamination with other species[17], where isolates with > 90% assigned to *P. malariae* were kept for further analysis, leaving a genomics dataset (using Illumina short read sequencing) of 153 isolates spanning 26 countries (n: Africa 128, Asia 18, South America 6, Oceania

1), with a total of 131,601 unique SNP positions. For long-read sequencing ONT data generated from the Bangladesh isolates ($n = 4$), basecalling was performed using Guppy (v6.1) with the MODEL SWAP configuration. PyoQC software was used to assess the quality of the sequencing data, and filtering was performed using the FiltLong (v0.2.1) tool. Variants were called on the filtered sequence data using Clair3 software, and merged with the Illumina-based data, leading to a final dataset for population genetics analysis consisting of 157 isolates (Supplementary data 1, 2, 3; Asia $n = 22$; 28 countries).

When investigating SNPs within genes associated with drug susceptibility, we used a semi-filtered database where the unfiltered database was filtered to only include isolates with a centrifuge score > 90% for *P. malariae*. The semi-filtered databased included 194 isolates within Central Africa ($n = 39$), East and Southern Africa ($n = 62$), West Africa ($n = 60$), Asia ($n = 24$), Oceania ($n = 1$) and South America ($n = 8$) and allowed for detection of low frequency mutations.

### Population genetics
Isolates were grouped at the both the continent and regional level and assessed for multiclonality using the MOIMIX package (v0.0.2.9001) to calculate the $F_{WS}$ score[25]. EstMOI software[68] was also used to investigate multiplicity of infection, which correlated negatively with $F_{WS}$ scores (Spearman's rho = −0.66). Only biallelic SNPs within the core genome were used in $F_{WS}$ calculations. Isolates with an $F_{WS}$ score > 0.85 ($n = 136$) (Supplementary data 1, 2, 3) were designated as monoclonal for use in robust population genetic analysis, specifically for homology, IBD and selection analyses. Other analyses using the filtered genomics database of 157 isolates (not filtered for monoclonal isolates) included population structure using principal component analyses (PCA) calculated with the QQMAN package[69], and maximum likelihood trees constructed with IQTREE software (GTR + F + R4 substitution model as assigned with ModelFinder; 1,000 bootstrap replicates)[70]. SNP positions driving differences in allele frequencies between populations were identified using the fixation index ($F_{ST}$), calculated using the VCFTOOLS package[66]. The ADMIXTURE package (v1.3.0) was used to perform ancestry analysis in all filtered isolates, for countries that had >5 isolates per country ($n = 141$ isolates), with the most likely number of ancestral populations estimated using the cross-validation error score (as calculated from an average of 10 replicates)[71]. Signals of positive selection within the monoclonal genomic dataset ($n = 126$) were calculated using the REHH R package (v3.2.1)[31]. The integrated haplotype score (iHS) was calculated within population and the Rsb score was calculated to identify signals of selection comparing two different populations. Both scores were calculated using monoclonal isolates ($F_{WS} > 0.85$) with groups of >10 isolates. Signals of homology at the country and continent level were investigated through screening for identity-by-descent (IBD) using the hmmIBD package (v2.0.4; default parameters)[72]. South American isolates were excluded from iHS and Rsb analyses due to small population sizes.

### In silico protein modelling for genes of interest
Amino acid sequences for all 20 genes of interest were obtained from PlasmoDB[73] using the PmUG01 reference genome for *P. malariae* (PmUG01_05034700) and the 3D7 reference genome for *P. falciparum* (PF3D7_0417200). ClustalO[74] software was used to align the respective *P. malariae* and *P. falciparum* orthologs to each other, and the resulting alignments were visualised and manually annotated using the JalView[75] programme.

The crystal structure of PfDHFR complexed with NADPH, dUMP and pyrimethamine (3QGT) was obtained from the RCSB protein data bank (https://www.rcsb.org/structure/3QGT). The structure of PmDHFR was predicted using iTASSER[76], with the amino acid sequence for PmUG01_05034700 obtained from PlasmoDB. Both protein structures were visualised and aligned in UCSF Chimera, with manual annotation[77].

## Molecular cloning: generating pDonor and pCas9 plasmids

Co-transfection of two plasmids was used to generate transfectant parasite lines: 1) a Cas9 encoding plasmid with the 20 bp guide sequence (5'-ccaagtacgagaagttaaag-3') targeting *pkdhfr*, pCas9_sgDHFR and 2) individual donor plasmids containing the DHFR domain sequence to replace PkDHFR with, flanked by two 500 bp homology regions. The 20 bp guide sequence targeting *pkdhfr* was integrated into pCas9_sg using infusion ligation as previously described[57], with sequences in Supplementary data 12. Individual donor plasmids containing the DHFR domain for ortholog replacement flanked by restriction enzymes were ordered from GeneART. One donor plasmid, containing the pkdhfr[OR] domain was flanked with two restriction enzymes (SpeI and NcoI, NEB) and the two 500 bp homology regions. Homology regions were added to the other ortholog replacement domains using restriction digests and ligation with T4 DNA Ligase as per manufacturer's instructions (NEB).

## Parasite culture and transfection

*P. knowlesi* parasites were cultured in human blood donated by the United Kingdom National Blood Transfusion Service after confirmation of Duffy-positive (Fy + ) blood status following previously published guidelines[56]. Parasites were grown in complete medium, comprised of RPMI 1460 (Invitrogen) with additions including: 2.0 g/L sodium bicarbonate, an additional 2.0 g/L D-glucose, 25 mM HEPES, 0.05 g/L hypoxanthine, 5 g/L AlbuMAX II, 0.0025 g/L gentamicin sulphate, 2 mM L-glutamine and 10% (vol/vol) horse serum (Pan Biotech; P30-0702) according to previously published protocols[56,57]. Gradient centrifugation with 55% Nycodenz (Progen; product 1002424) in RPMI was used to synchronise *P. knowlesi* parasites via enriching schizonts, as previously described with an incubation with 1 μM Compound 2 (PKG/egress inhibitor) for 2–3 h to tighten synchronisation[57].

For each transfection, wild-type *P. knowlesi* A1-H.1 parasites were co-transfected with two plasmids: 1) Cas9-encoding plasmid directed to the *pkdhfr* locus via the guide RNA sequence, including a drug selectable marker (blasticidin resistance cassette, BSDr) and 2) the individual donor plasmids containing the DHFR domain ortholog to be replaced flanked by 500 bp homology regions either side of the *pkdhfr* locus to allow for homology directed repair (HDR). The CRISPR-Cas9 plasmid encoding the Cas9 endonuclease and sgRNA cassette *pCas9_sg*[78] was modified to replace the hDHFR-yFCU selection cassette with a BSDr-yFCU selection cassette (BSDr-yFCU, modified from Patel et al.[79]) to negate the need to use antifolates for selection of transgenic *P. knowlesi* parasites. *P. knowlesi* parasites are less sensitive to BSD than *P. falciparum*[59], so a higher than standard concentration of 25 μg/mL was used for selection in comparison to 5 mg/mL commonly used for positive selection with *P. falciparum*[79]. Previous work using an endogenously expressed BSDr demonstrated that unmodified parasites had an IC50 of 20 mM and modified parasites demonstrated an IC50 of 30 mM[58].

Tightly synchronised schizonts were transfected using the Amaxa 4D electroporator (Lonza) with 100 μL P3 Primary Cell 4D Nucleofector X Kit L (Lonza) using 10 μl mixed DNA containing >10 μg Cas9-encoding plasmid and >10 μg and 10 μL packed schizonts[57,78]. Following transfection, schizonts were transferred to a 6-well plate and 4.5 mL media to create a 5 mL total culture with 4% haematocrit and grown under normal parasite conditions as described above. After 27 hours, the media is replaced with parasite media containing 25 μg/mL positive selection drug, BSD and media for 5 days following transfection. Successful transfectant parasites were confirmed by diagnostic PCR as previously described[57]. Genomic DNA was extracted using the Blood genomicPrep Mini spin kit (Cytiva) and amplified using CloneAmp HiFi PCR Premix (TakaraBio) using 35 cycles of the following conditions: 98 °C for 10 sec, 55 °C for 15 sec, 72 °C for 5 sec/kb of product. Primers for genomic confirmation of transfectant clones, in addition to the guide sequence targeting Cas9 to *pkdhfr* are provided in Supplementary data 12.

## Parasite growth rate assays

A flow cytometry-based assay was used to measure parasite growth over one asexual cycle (27 h) as previously described[78]. For all parasite lines, schizonts were purified using Nycodenz and Compound 2. Purified schizonts were then re-suspended in 2% haematocrit culture at ~0.5% parasitemia in triplicate technical repeats. Parasitaemia was measured using a flow cytometry assay at 0 h, after the cultures were plated but before incubation, and at 27 h after they had been incubated at 37 °C. For the flow cytometry readings, live parasite cultures were stained with SYBR Green I (Life Technologies) before analysis in the Attune Cytometric Software (v6.0). Data was normalised to a 1% starting parasitemia to enable comparison of fold multiplication between lines. Experiments were all carried out with triplicate biological independent replicates, with each biological replicate an average of three technical replicates.

## Parasite growth inhibition assay with drugs

Chloroquine, pyrimethamine and dihydroartemisinin (DHA) were supplied by the Medicines for Malaria Venture, Geneva, Switzerland. Chloroquine was used as a negative control to confirm parasite death, and dihydroartemisinin was used in parallel with pyrimethamine as a control drug with a differing mode of action. Chloroquine stocks were prepared in distilled water, and both pyrimethamine and DHA stocks were prepared in dimethyl sulfoxide (DMSO).

Drug susceptibility assays were set up individually for all parasite lines as previously described[59] with parasites exposed in triplicate technical repeats to serial dilutions of DHA and pyrimethamine for 27 h. After 27 h, parasites were stained with SYBR green I (Thermo Fisher Scientific; product S7563) to assess for viability, with fluorescence analysed using the Spectramax M3 plate reader (Molecular Devices) with excitation at 490 nm and reading at 520 nm.

## URLs

R Statistical software, https://www.r-project.org.
JalView software, https://www.jalview.org.

## Reporting summary

Further information on research design is available in the Nature Portfolio Reporting Summary linked to this article.

## Data availability

All novel raw sequencing data is available under study accession PRJEB75553, with individual isolate accession codes in supplementary data 2. All previously sequenced isolates are listed in supplementary data 2 with the accession codes used to download this data and the respective research papers referenced in the methods.

## Code availability

For sequence data analysis, please see the Malaria Hub Github repository (https://github.com/LSHTMPathogenSeqLab/malaria-hub). All analysis was performed using open-source software named and referenced in the methods.

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

## Acknowledgements

We thank Ernest Benavente for bioinformatics advice on an earlier version of the analysis. T.G.C. and S.C. are funded by UKRI MRC (MRC IAA2129 T.G.C, MR/R026297/1 T.G.C, and MR/X005895/1 T.G.C) and EPSRC (EP/Y018842/1 T.G.C), and Wellcome iTPA Translational Accelerator Award (214227/Z/18/Z T.G.C) grants. R.W.M and F.M. were supported by an MRC Career Development Award (MR/M021157/1 R.W.M) jointly funded by the U.K. Medical Research Council and Department for International Development. R.W.M and A.I are also funded by a Wellcome Trust Discovery Award (225844/Z/22/Z R.W.M). We thank the collaborating teams in Bangladesh, Brazil, Gabon, and Thailand for sharing *P. malariae* isolates for this project. We thank Terence Boussougou-Sambe, Barclaye Ngossanga, Tatiana Röckl-Pinilla, Sara Buezo Montero, Andrea Weierich, Anton Hoffmann, Maxim Viehweg, and Sarah Gräßle. We also thank the patients and volunteers (adults, children, and their parents) for participating in this study and donating venous blood samples.

## Author contributions

R.W.M., T.G.C., and S.C. conceived and directed the project. A.I., D.N., S.B., A.A.A., S.M.D.S., M.S.A, D.M., F.N., C.J.S. and S.C. coordinated sample collection. A.I., D.N., S.B., and S.C undertook DNA extraction.

A.I. and S.C. coordinated whole genome sequencing. A.I., E.M., and J.E.P. performed bioinformatic and statistical analyses under the supervision of T.G.C. and S.C. The laboratory validation work using the *P. knowlesi* culture system was performed by A.I., F.M., and D.A.v.S. under the guidance of R.W.M. and S.C. A.I., R.W.M., T.G.C., and S.C. interpreted results. A.I., R.W.M., T.G.C., and S.C. wrote the first draft of the manuscript. All authors commented on and edited various versions of the draft manuscript.

## Competing interests

The authors declare no competing interests.
