## [Peer Review file · Nature Communications]

Genome sequencing of *Plasmodium malariae* identifies continental segregation and mutations associated with reduced pyrimethamine susceptibility

Corresponding Author: Dr Amy Ibrahim

Version 1:

Reviewer comments:

Reviewer #1

(Remarks to the Author)

This study by Ibrahim et al explores the natural variation observed in the malaria parasite *Plasmodium malariae*, using whole genome sequencing of patient isolates. *P. malariae* is something of a neglected *Plasmodium* in that it is less pathogenic than other human malaria parasites and is unable to be cultured in the lab, limiting experimental investigation. This study moves the field forward in that regard, building on the authors work using the tractable *P. knowlesi* species to engineer genetic variants for phenotypic analysis. The authors choose dhfr, the target of pyrimethamine, to examine in detail as *P. malariae* polymorphisms coincide with known resistance mutation in *P. falciparum* dhfr. Notably, they observe that the anticipated resistance SNPs do indeed shift tolerance to pyrimethamine, and also that the reference sequence for *malariae* includes two SNPs conferring resistance to pyrimethamine.

The authors also highlight several other candidate genes (including drug resistance, vaccine target, RBC invasion) for which several polymorphisms are detected across isolates. However the functional impact of these is unclear and awaits future experimentation. Of particular interest would be those for which exposure to current treatments might be expected to exert selection pressure.

The value in the analysis presented here of the 251 *P. malariae* genomes is also as a benchmark for future sequencing studies to understand how parasite genomes are evolving over time, particularly as new therapeutic options begin to be deployed.

There are a few areas where the manuscript could be clarified, noted below, with only minor textual changes suggested.

I may have missed it, but where does the reference strain originate and how does it cluster with the other isolates? Some comment on when and where the current reference genome was isolated would be of interest, particularly given the pre-existing pyrimethamine resistance mutations noted.

The supplementary tables are a bit hard to navigate as they lack titles or a description – some additional information would be valuable.

The authors validate BSD as a *P. knowlesi* selection marker for the first time here, and show there are no differences in parasite growth in lines expressing this marker. It would also be useful for the community to show how the drug EC50 values compare between untransfected and modified parasite lines to guide the selection window.

Reviewer #2

(Remarks to the Author)

Ibrahim and colleagues present a very interesting study of *Plasmodium malariae* genomics. The CRISPR knockout in *Plasmodium knowlesi* is particularly groundbreaking and provides evidence that orthologs of genes associated with antimalarial resistance in *Plasmodium falciparum* have this same association in a nonculturable species. The authors

correctly identify this work as their major finding, and these results are convincing. The global genomic analysis is weaker, but nonetheless interesting and warrants publication.

Major comments:

1. The reduced pyrimethamine susceptibility in the CRISPR-edited Pk is undoubtedly an important finding, but the context and potential real-world consequences of this work should be more fully developed to strengthen the manuscript. In particular, I would encourage the authors to put these results in the context of pyrimethamine drug pressure on Pm populations. How widely is SP used in the countries where samples originated? Is there likely to be significant drug pressure in the countries where mutations were identified? It might also be worth addressing the likelihood of pyrimethamine resistance in Pm being driven by treatment for Pf, given that Pm is rarely screened for and causes persistent infections. I would also suggest adding some discussion of delayed clearance in Pf to strengthen the case that these Pm mutants may be clinically meaningful.

2. The authors have wrangled an impressive number of Pm isolates, including from the largely unstudied Asian and South American populations. Unfortunately, the genomic analysis lacks the heft and rigor of the CRISPR knockout work. Specific comments follow:

a. I am not convinced there is a meaningful geographic separation between East/South Africa and West/Central Africa. Even in the supplemental PC2 where the authors claim it exists, there are numerous outliers. If this separation is clearly borne out in a discriminant PCA, I could be swayed, but as is, I don't buy it.

b. The ADMIXTURE results are not presented well. If $K=2$ had the lowest cross-validation error (and it does from what I can see), why are the graphs showing $K=4$? There may be clearer separation but it doesn't align with the statistical evidence that suggests there are only two distinct populations. If the authors wish to present the data this way, it should be clearly indicated that it is for ease of visualization only. If the maximum likelihood phylogeny showed 4 populations, you could make an argument for believing that instead of ADMIXTURE, but as far as I can tell, it also shows only 2.

In addition, it is only using $K=4$ that there's any evidence for South American Pm having a mixed African/Asian ancestry. In the $K=2$ supplemental plot, this does not appear to be true; rather, the South American population appears to be derived from the African population (which also makes much more sense given historical migration of humans and their associated Plasmodium parasites).

In general, I would advise caution about overinterpreting ADMIXTURE data that shows mixed ancestry. My interpretation of your results is that 2 populations are most likely, and that the Asian population derives from the African one, which is both an interesting finding and in alignment with previous studies that suggest Pm is derived from an African ape Plasmodium.

c. I would strongly prefer COI/MOI to Fws both due to the arbitrary nature of Fws cutoffs and the ease of interpretation. I suspect you will find mostly monoclonal samples as with Fws so unlikely to change the results in a major way.

d. The IBD segments part of the IBD analysis is great and provides some interesting areas for further investigation. The way that the authors present pairwise IBD, however, is strange to me and difficult to interpret, particularly within the African samples. It would be more informative to divide isolate pairs into bins, e.g. $IBD > 0.25$, $IBD > 0.5$, $IBD > 0.9$, etc. I recognize that this isn't the main thrust of the paper, but some IBD networks might also be illuminating and would show that some African isolates are highly interrelated (my inference based on the outliers in Fig SF5).

Minor comments:

1. The authors refer to "difficulties in diagnosing infections" in the second paragraph of the introduction. What are those difficulties?

2. This is a small nitpick, but Pm can cause fairly high density infections especially compared to e.g. *P. ovale*. So it strikes me that obtaining sufficient DNA from Pm infections isn't as much of a limitation as the authors claim at the end of the introduction.

3. Which Congo did your samples come from?

4. Removing mixed infections is justifiable and cautious, but what was the rationale for not filtering out non-Pm reads and using the samples anyway?

5. In the "Geographically variant residues..." section of the results, the authors present evidence from Pf and *P. berghei*. It should be more explicitly stated that all of this evidence comes from orthologs and may not necessarily have any clinical importance in Pm (especially when the data comes from *berghei*).

6. See my above comment about using Fws, but is it actually lower in Africa/SAM than in Asia? There's no statistical comparison given. COI would make this clearer too.

7. See my above comment about IBD. I don't think the way the text is currently presented on this is very clear, especially given the huge number of outliers in the African isolates.

8. The iHS section of the results (page 6-7) is presented strangely. My interpretation of the authors' intent is to point to genomic differentiation between the African and Asian populations. This paragraph should be rewritten with that in mind, e.g. "Scans for positive selection identified different genomic regions under selective pressure between the African and Asian populations"

9. The claim on page 11 that Pm causes "untraceable transmission that can drive malaria recurrence" needs a citation.

10. The claim at the end of the first full paragraph on page 12 that Pm may geographically differentiate with more sample size strikes me as unlikely given the actual evidence. Not being able to differentiate Mali and Burkina Faso in Pf is quite different from not being able to reliably differentiate East and West Africa! In addition, smaller sample sizes than this have shown geographic separation in other Plasmodium species (e.g. Po).

11. I am unable to identify any isolates from South America in the phylogeny in Figure 1A. The Oceania branch is difficult to see as well.

12. How is the PCA in Figure 1B different from the one in Figure SF2?

13. See my above comment about ADMIXTURE re: Figure 1C.

Version 2:

Reviewer comments:

Reviewer #1

(Remarks to the Author)

The authors have addressed all my comments in this revision, and added additional textual changes that improved clarity..

The only minor note is that the title for Suppl. Table 4 is mislabelled as Suppl. Table 3.

Reviewer #2

(Remarks to the Author)

The authors have thoughtfully addressed my comments from the first version of the manuscript in this revised version. I have no further comments.

REBUTTAL

Reviewer #1 (Remarks to the Author):

This study by Ibrahim et al explores the natural variation observed in the malaria parasite *Plasmodium malariae*, using whole genome sequencing of patient isolates. *P. malariae* is something of a neglected *Plasmodium* in that it is less pathogenic than other human malaria parasites and is unable to be cultured in the lab, limiting experimental investigation. This study moves the field forward in that regard, building on the authors work using the tractable *P. knowlesi* species to engineer genetic variants for phenotypic analysis. The authors choose dhfr, the target of pyrimethamine, to examine in detail as *P. malariae* polymorphisms coincide with known resistance mutation in *P. falciparum* dhfr. Notably, they observe that the anticipated resistance SNPs do indeed shift tolerance to pyrimethamine, and also that the reference sequence for *malariae* includes two SNPs conferring resistance to pyrimethamine. The authors also highlight several other candidate genes (including drug resistance, vaccine target, RBC invasion) for which several polymorphisms are detected across isolates. However the functional impact of these is unclear and awaits future experimentation. Of particular interest would be those for which exposure to current treatments might be expected to exert selection pressure. The value in the analysis presented here of the 251 *P. malariae* genomes is also as a benchmark for future sequencing studies to understand how parasite genomes are evolving over time, particularly as new therapeutic options begin to be deployed. There are a few areas where the manuscript could be clarified, noted below, with only minor textual changes suggested.

I may have missed it, but where does the reference strain originate and how does it cluster with the other isolates? Some comment on when and where the current reference genome was isolated would be of interest, particularly given the pre-existing pyrimethamine resistance mutations noted.

Thank you for your positive comments. There is now more detail on the reference genome (Methods).

The supplementary tables are a bit hard to navigate as they lack titles or a description – some additional information would be valuable.

We agree, and have added titles and descriptions in the Supplementary tables.

The authors validate BSD as a *P. knowlesi* selection marker for the first time here and show there are no differences in parasite growth in lines expressing this marker. It would also be useful for the community to show how the drug EC50 values compare between untransfected and modified parasite lines to guide the selection window.

A previous study (10.1093/jac/dkx279) identified the EC50 of PkA1H1 (the un-transfected and unmodified parental parasite line for this study) to Blasticidin as 31684 nM, in comparison to P. falciparum EC50 of 1413 nM (22-fold difference). This is referenced and this study guided us to use a BSD concentration of 25 ug/mL for selection of successfully transfected parasite lines as an increased concentration in comparison to 5 ug/mL commonly used in P. falciparum positive selection. We used a 5-fold increase in BSD concentration in comparison to P. falciparum positive selection.

Additionally, previous work (<https://doi.org/10.1016/j.molbiopara.2003.10.019>) had validated Blasticidin-S as a positive selection marker for genetic modification in P. knowlesi. In this study, wild type, unmodified P. knowlesi lines demonstrated an IC50 of 20 mM for Blasticidin-S, and modified lines which contained an episomal plasmid with a Blasticidin resistance cassette demonstrated an IC50 of 30 mM (3.1-fold decrease in susceptibility due to the BSD resistance cassette). We have added a mention of this in the Results section with the reference to aid other researchers when using BSD as a positive selection marker in P. knowlesi research.

Reviewer #2 (Remarks to the Author):

Ibrahim and colleagues present a very interesting study of *Plasmodium malariae* genomics. The CRISPR knockout in *Plasmodium knowlesi* is particularly groundbreaking and provides evidence that orthologs of

genes associated with antimalarial resistance in *Plasmodium falciparum* have this same association in a nonculturable species. The authors correctly identify this work as their major finding, and these results are convincing. The global genomic analysis is weaker, but nonetheless interesting and warrants publication.

Thank you for highlighting the strength of the functional work.

Major comments:

1. The reduced pyrimethamine susceptibility in the CRISPR-edited Pk is undoubtedly an important finding, but the context and potential real-world consequences of this work should be more fully developed to strengthen the manuscript. In particular, I would encourage the authors to put these results in the context of pyrimethamine drug pressure on Pm populations.

How widely is SP used in the countries where samples originated? Is there likely to be significant drug pressure in the countries where mutations were identified?

We have added more information in the Discussion section on current and historical SP usage specifically in Africa, where SP is still used as preventative therapy, and resistance mutations are seen at the highest prevalence.

It might also be worth addressing the likelihood of pyrimethamine resistance in Pm being driven by treatment for Pf, given that Pm is rarely screened for and causes persistent infections.

We have added text in the Discussion addressing that Pf treatment is likely driving selection in Pm.

I would also suggest adding some discussion of delayed clearance in Pf to strengthen the case that these Pm mutants may be clinically meaningful.

Good idea. In the Discussion, we have included a sentence on the potential clinical implications of the identified mutations in Pm isolates. Additionally, we have incorporated our finding of the PmDHFR F57L variant and discussed its potential relevance. Identifying this variant in Pm is particularly noteworthy, as there is currently no clinical threat posed by an orthologous variant in Pf. This suggests an opportunity to monitor for the emergence of this orthologous variant in Pf before it becomes a clinically significant concern.

2. The authors have wrangled an impressive number of Pm isolates, including from the largely unstudied Asian and South American populations. Unfortunately, the genomic analysis lacks the heft and rigor of the CRISPR knockout work. Specific comments follow:

a. I am not convinced there is a meaningful geographic separation between East/South Africa and West/Central Africa. Even in the supplemental PC2 where the authors claim it exists, there are numerous outliers. If this separation is clearly borne out in a discriminant PCA, I could be swayed, but as is, I don't buy it.

Yes, we agree this is not clear-cut, particularly as there are numerous outliers. We have amended the Results section to not overinterpret the PCA, and to reflect that there may be some separation, but this is currently not a clear population structure, and sequencing of greater isolates may be needed to decipher this.

b. The ADMIXTURE results are not presented well. If K=2 had the lowest cross-validation error (and it does from what I can see), why are the graphs showing K=4? There may be clearer separation but it doesn't align with the statistical evidence that suggests there are only two distinct populations. If the authors wish to present the data this way, it should be clearly indicated that it is for ease of visualization only. If the maximum likelihood phylogeny showed 4 populations, you could make an argument for believing that instead of ADMIXTURE, but as far as I can tell, it also shows only 2.

In addition, it is only using K=4 that there's any evidence for South American Pm having a mixed African/Asian ancestry. In the K=2 supplemental plot, this does not appear to be true; rather, the South American population appears to be derived from the African population (which also makes much more sense given historical migration of humans and their associated *Plasmodium* parasites)

In general, I would advise caution about overinterpreting ADMIXTURE data that shows mixed ancestry. My interpretation of your results is that 2 populations are most likely, and that the Asian population derives from the African one, which is both an interesting finding and in alignment with previous studies that suggest Pm is derived from an African ape Plasmodium.

Thank you for the insightful comment. The K=2 population structure is statistically most likely, therefore we have amended this and included K=2 in the main figure (Figure 1C) and we have added text in the Results section to demonstrate that K=2 is the most likely value of populations, with K = 4 included to show suggestions of population structure within Africa, and for visualisation of this only.

c. I would strongly prefer COI/MOI to Fws both due to the arbitrary nature of Fws cutoffs and the ease of interpretation. I suspect you will find mostly monoclonal samples as with Fws so unlikely to change the results in a major way.

We also ran estMOI and found that the results agreed with FWS. However, we chose to present FWS data because it is commonly presented. We have added information about estMOI in the Methods.

d. The IBD segments part of the IBD analysis is great and provides some interesting areas for further investigation. The way that the authors present pairwise IBD, however, is strange to me and difficult to interpret, particularly within the African samples. It would be more informative to divide isolate pairs into bins, e.g. IBD>0.25, IBD>0.5, IBD>0.9, etc. I recognize that this isn't the main thrust of the paper, but some IBD networks might also be illuminating and would show that some African isolates are highly interrelated (my inference based on the outliers in Fig SF5).

We agree. The large number of outliers make it difficult to interpret the boxplot. As suggested, we have binned the pairwise scores (see ST5), demonstrating that African isolates have most segments in the 0 – 0.05 bin, whereas Asian pairwise scores are highest in the 0.05 – 0.1 bin, suggesting a potential difference in inter-relatedness between parasites in these two regions (consistent with the boxplot). We have updated the Results section.

Minor comments:

1. The authors refer to "difficulties in diagnosing infections" in the second paragraph of the introduction. What are those difficulties?

These difficulties were mentioned in paragraph 1, where we have now expanded on them.

2. This is a small nitpick, but Pm can cause fairly high density infections especially compared to e.g. P. ovale. So it strikes me that obtaining sufficient DNA from Pm infections isn't as much of a limitation as the authors claim at the end of the introduction.

Pm may cause higher parasitemia infections than P. ovale. The final sentence is referring to all neglected Plasmodium species including P. ovale species. We have tried to sequence clinical isolates of Pm isolates without SWGA, and had difficulties obtaining useable data, so we believe that the parasitaemia and amount of parasite material obtained from Pm infections is a hindrance to genome studies.

3. Which Congo did your samples come from?

Eight isolates sequenced were from Congo (PM_COG_001 – PM_COG_008) and one isolate was from the Democratic Republic of Congo (PM_COD_001). This is now shown in ST2 and ST1.

4. Removing mixed infections is justifiable and cautious, but what was the rationale for not filtering out non-Pm reads and using the samples anyway?

We are sequencing isolates using short-read (250bp) sequencing methods. Some genes within the Plasmodium genomes are highly conserved, and therefore reads corresponding to another Plasmodium spp. may map with the Pm reference genome, even though they are not genuine Pm reads, leading to potentially miscalled SNPs in Pm. Therefore, we chose to not include these isolates. There are ways to avoid mis-mapping of sequence data, including creating a Pan-Plasmodium genome, and extracting only reads mapping to Pm. However, this novel approach

needs to be tested on sample sets of mixed infections to validate it as a robust methodology. In lieu of such work, we decided to select only Pm mono-infections that have a clear interpretation.

5. In the "Geographically variant residues..." section of the results, the authors present evidence from Pf and P. berghei. It should be more explicitly stated that all of this evidence comes from orthologs and may not necessarily have any clinical importance in Pm (especially when the data comes from berghei).

We agree, and have amended the text, and added a sentence to reflect this.

6. See my above comment about using Fws, but is it actually lower in Africa/SAm than in Asia? There's no statistical comparison given. COI would make this clearer too.

We have used a Wilcoxon test to compare the difference between Fws scores, confirming a difference between African and Asian populations (see Results).

7. See my above comment about IBD. I don't think the way the text is currently presented on this is very clear, especially given the huge number of outliers in the African isolates.

We agree. The large number of outliers make it difficult to interpret the boxplot. We have binned the pairwise scores (see ST5) and have edited the interpretation of the results in the text.

8. The iHS section of the results (page 6-7) is presented strangely. My interpretation of the authors' intent is to point to genomic differentiation between the African and Asian populations. This paragraph should be rewritten with that in mind, e.g. "Scans for positive selection identified different genomic regions under selective pressure between the African and Asian populations"

Sorry for the confusion. The iHS score is calculated in 1 population only (e.g., signals within Africa) and is not compared to any other population, whereas the Rsb score is calculated by comparing the signals between African and Asian isolates. Therefore, they are providing two slightly different results, which are commonly presented together. We have edited this section to better reflect this.

9. The claim on page 11 that Pm causes "untraceable transmission that can drive malaria recurrence" needs a citation.

Good point. We have added a reference.

10. The claim at the end of the first full paragraph on page 12 that Pm may geographically differentiate with more sample size strikes me as unlikely given the actual evidence. Not being able to differentiate Mali and Burkina Faso in Pf is quite different from not being able to reliably differentiate East and West Africa! In addition, smaller sample sizes than this have shown geographic separation in other Plasmodium species (e.g. Po).

Agreed. It is likely that this is an interesting unique characteristic of Pm, and the text is now updated.

11. I am unable to identify any isolates from South America in the phylogeny in Figure 1A. The Oceania branch is difficult to see as well.

The branches are coloured to reflect the continent of origin of an isolate, and this branch/tip colour may be difficult to see where the isolates share a clade with an isolate from a different continent. The outer circle track indicates the region of origin of every isolate. We have edited the figure legend to make this clearer, and the image uploaded for publication will be of high resolution.

12. How is the PCA in Figure 1B different from the one in Figure SF2?

Figure 1B includes all isolates in all continents in the filtered database. Whereas, SF2 is presented by continent, with SF2A showing only Asian isolates and SF2B showing only African isolates. The text and figure legends are now clearer.

13. See my above comment about ADMIXTURE re: Figure 1C.

The admixture results interpretation and figures have been amended to show clearly that K=2 is the most likely number of ancestral populations, with K=4 included only to suggest preliminary population structure in Africa, whilst it is currently not statistically supported. Further sequencing of

Pm isolates from Africa could either aid to demonstrate population structure within the African continent, or confirm that within Africa, there is no clear segregation of Pm parasites.